# Tumor-Infiltrating Lymphocytes (TILs) in Epithelial Ovarian Cancer: Heterogeneity, Prognostic Impact, and Relationship with Immune Checkpoints

**DOI:** 10.3390/cancers14215332

**Published:** 2022-10-29

**Authors:** Delphine Hudry, Solenn Le Guellec, Samuel Meignan, Stéphanie Bécourt, Camille Pasquesoone, Houssein El Hajj, Carlos Martínez-Gómez, Éric Leblanc, Fabrice Narducci, Sylvain Ladoire

**Affiliations:** 1Inserm, U1192–Protéomique Réponse Inflammatoire Spectrométrie de Masse–PRISM, Lille University, F-59000 Lille, France; 2Department of Gynecologic Oncology, Oscar Lambret Center, F-59000 Lille, France; 3Tumorigenesis and Resistance to Treatment Unit, Centre Oscar Lambret, F-59000 Lille, France; 4CNRS, Inserm, CHU Lille, UMR9020-U1277-CANTHER-Cancer Heterogeneity Plasticity and Resistance to Therapies, Lille University, F-59000 Lille, France; 5Department of Medical Oncology, Centre Georges-François Leclerc, F-21000 Dijon, France; 6INSERM, CRI-866 Faculty of Medicine, F-21000 Dijon, France

**Keywords:** epithelial ovarian cancer (EOC), high-grade serous ovarian cancer (HGSOC), tumor-infiltrating lymphocytes (TILs), immune checkpoint inhibitors, peritoneum, tumor microenvironment

## Abstract

**Simple Summary:**

Outcomes of ovarian cancer (OC) patients remain poor despite recent advances in oncology. Immunotherapy has proven to be beneficial in treating selected populations with different cancer types. However, OC shows very little response to immunotherapy; thus, it is mandatory to understand which subgroups of OC patients might benefit the most and how to potentiate its effect. In recent years, a more comprehensive understanding of the immune microenvironment of OC has been described, especially regarding the characterization of tumor-infiltrating lymphocytes (TILs). These results are promising and open the fields to new therapeutic approaches incorporating immune checkpoint blockade. This review aims to synthesize recent research on TILs in OC patients.

**Abstract:**

Epithelial ovarian cancers (EOC) are often diagnosed at an advanced stage with carcinomatosis and a poor prognosis. First-line treatment is based on a chemotherapy regimen combining a platinum-based drug and a taxane-based drug along with surgery. More than half of the patients will have concern about a recurrence. To improve the outcomes, new therapeutics are needed, and diverse strategies, such as immunotherapy, are currently being tested in EOC. To better understand the global immune contexture in EOC, several studies have been performed to decipher the landscape of tumor-infiltrating lymphocytes (TILs). CD8+ TILs are usually considered effective antitumor immune effectors that immune checkpoint inhibitors can potentially activate to reject tumor cells. To synthesize the knowledge of TILs in EOC, we conducted a review of studies published in MEDLINE or EMBASE in the last 10 years according to the PRISMA guidelines. The description and role of TILs in EOC prognosis are reviewed from the published data. The links between TILs, DNA repair deficiency, and ICs have been studied. Finally, this review describes the role of TILs in future immunotherapy for EOC.

## 1. Introduction

Despite advances in medical treatment and surgery, ovarian cancer is the most lethal gynecological cancer. According to recent epidemiological estimates, there were 21,410 news cases of ovarian cancers in the United States in 2021, accounting for 13,770 deaths; this makes it the sixth most common cause of cancer-related death in women after lung, breast, colon, rectum, and pancreas cancers [1], with a 5-year survival rate of 46% [2] when all stages are combined. The most common histologic type is epithelial ovarian cancer (EOC), with 75% of the patients being diagnosed at advanced FIGO stages (III or IV) [2]. EOC includes a wide range of diseases with varying prognoses. The most common histologic subtypes are the following: high-grade serous ovarian carcinoma (HGSOC), which is usually associated with a poor prognosis; low-grade serous carcinoma (LGSOC); mucinous carcinoma; endometroid carcinoma; and clear cell carcinoma (CCC) [3]. These different subtypes present distinct anatomical origins, molecular profiles, and prognoses. The recommended first-line therapy includes complete cytoreduction associated with carboplatin–paclitaxel intravenous chemotherapy [4]. Three randomized controlled trials [5,6,7] showed that, in patients with a high tumor burden who are eligible for primary cytoreduction surgery (pCRS), neoadjuvant chemotherapy followed by interval cytoreduction surgery (iCRS) is associated with lower surgery-related morbidity. However, retrospective studies highlight the contrast between pCRS and iCRS patients’ profiles [8]. Patients who undergo iCRS usually present a high tumor burden that prevents upfront surgery. Patient evaluation with imaging and diagnostic laparoscopy is essential to determine the patient’s eligibility for pCRS [9]. The standard chemotherapy regimen consists of three-week cycles of carboplatin at AUC 5/6 and paclitaxel at 175 mg/m^2^ [10]. The number of cycles is determined by the tumor load, the disease stage, the presence of metastases, and the patient’s condition. In recent years, new treatment options, such as hyperthermic intraperitoneal chemotherapy (HIPEC) following iCRS [11,12], have been shown to improve overall survival (OS) and disease-free survival (DFS). Currently, studies are evaluating the impact of HIPEC during pCRS [13,14]. Adding bevacizumab, an antiangiogenic drug (vascular endothelial growth factor (VEGF) inhibitor), to chemotherapy or using it as maintenance therapy has been shown to result in an improvement in DFS in women with a high risk of disease progression [15,16]. Poly(-adenosine diphosphate ribose) polymerase (PARP) inhibitors have also been shown to have a beneficial effect when used in maintenance therapy [15,17]. The use of olaparib in maintenance therapy for HGSOC and endometrioid ovarian cancer with BRCA mutation decreases the risk of recurrence [18]. Since 2011, targeted immunotherapy drugs have opened new horizons for cancer treatment [19]. Immune checkpoint inhibitors (ICIs) are being developed and evaluated in various types of cancers [20]. The use of immunotherapy requires a better understanding of the different aspects of the tumor’s immune microenvironment. Recently, tumor immune infiltrates have been widely studied in EOC, particularly tumor-infiltrating lymphocytes (TILs).

This study aims to provide an overview of the scientific literature on TILs in EOC. TILs evaluation, prognostic impact, and relation with the new treatment options, PARP inhibitors and ICIs, are summarized in a systematic review.

## 2. Materials and Methods

This systematic literature review was conducted according to the Preferred Reporting Items for Systematic Reviews and Meta-Analyses PRISMA guidelines [21], using specific eligibility criteria. We reviewed all articles identified through MEDLINE (via OVID) and EMBASE between January 2010 and August 2021. The combinations of mesh terms related to TIL subsets and EOC used in the search strategy are available in Appendix A. We included original English articles related to research studies using human EOC tissue. We excluded reviews, letters, and editorials. Studies focusing exclusively on tumor-associated neutrophils, macrophages, myeloid-derived suppressor cells, and natural killer cells were also excluded. The outcomes analyzed were TIL description, survival impact, and TIL modification with standard or new treatment.

All articles were independently screened for eligibility by two authors (DH and SLG). The process of selection was performed in two steps: the first step was the selection of the articles based on the titles and the abstracts, and the second step consisted of selecting the articles based on an evaluation of the full text. Two reviewers extracted the data independently and any disagreements were discussed with a third reviewer. For the included studies, data on study design, study period, number of patients, sample used, method and cut-off used to identify TILs, and other outcomes were extracted.

## 3. Results

### 3.1. Study Selection

Through our systematic search, 1279 unique articles were identified, including 211 that underwent full-text evaluation. The results of the review process are presented in Figure 1. After excluding articles with a too high risk of bias, study population, follow-up, and measurement of outcomes were examined. Ultimately, 122 original studies were included in the final review. The articles were grouped according to the main outcomes.

### 3.2. TIL Definition

TILs in EOC are a subject that has gained a lot of interest in the last five years. In the published scientific data, TILs are evaluated using different methods, including genetic signature, count of TILs in hematoxylin and eosin (H&E) based pathological immunohistochemistry (IHC), and immunofluorescence (IF) [22]. The International Immuno-Oncology Biomarkers Working Group defined, in 2017, “intra-epithelial” (iTILs) as TILs present in the tumor and “stromal” (sTILs) as TILs that are present within 1 mm beneath the epithelial layer [23]. To evaluate the inflammatory infiltrate, sTILs and iTILs are expressed in percentages or median counts. In IHC, between three and 10 fields in stained slides are observed in x200 or x400 high-power fields (HPF). TILs in H&E, in IHC, or in IF with specific antibodies are analyzed with either an absolute count or a semi-quantitative cut-off. For example, Goode and al. divided iTILs CD8+ infiltration into four categories: 0, low: 1–2, median: 3–19, and high ≥ 20 [24]. Other authors concluded that TILs > 5 or 10 per HPF should define positive iTILs in EOC [25].Tumoral tissues and ascites from pCRS patients represent the main samples used in the studies [26]. The data showed variability in the immune infiltrate among the different tissue samples within the same patient (ovaries, omentum, and peritoneum) [27,28,29,30,31]. One study confirmed the feasibility of TIL evaluation in tumor samples that is performed using a 16-gauge needle biopsy [32]. Systematic tumor core biopsies can represent the immune microenvironment [33].The description of TILs in EOC uses various techniques, especially the cut-offs are extremely variable between the studies. To date, there is no consensus, apart from H&E which is not yet used in clinical routine, on the type of marker or the thresholds to identify TILs.

### 3.3. TIL Phenotypes

In H&E, TIL evaluation includes mononuclear infiltrate, lymphocytes, and plasma cells [23]. Sometimes TILs are spotted using CD3 marking [34]. Various subsets of T cells may be found in EOC: CD8+ T cytotoxic and CD4+ T helper lymphocytes are identified by either the molecules on their surface or the pattern of cytokines they produce. T helper CD4+ cells are divided into subtypes: Th1 cells that produce interleukin 2 (IL-2) and interferon INF-𝛾 (acting on CD8+ cells); Th2 cells that produce IL-4, 5, 6, 10, and 13 (humoral immunity) [35]; Th17 cells that produce IL-17; and T Follicular helper (TFH) that interacts with B lymphocytes [36]. Tumor-infiltrating B lymphocytes (B-TILs) have been shown to be present in several solid tumors, including EOC [37]. Regulatory T cells (TREG) produce cytokines with immunosuppressive activities, including IL-10 and TGFβ [36]. The expressions of FOXP3 and CD25 often identify TREG. All immune cells are detected in different locations within and around the tumor epithelium. EOC is a heterogeneous disease regarding TILs. In terms of histologic subtypes, HGSOC is studied in the majority of studies. LGSOC, mucinous carcinoma, endometrioid carcinoma, and CCC are also studied in most studies, while carcinosarcoma is studied in only a few cases. Table 1 shows the description of TILs in the articles reviewed here. Only recent studies with relevant samples and/or results are shown to improve readability. In those studies, the number of cases ranges from a few dozen patients to several hundred, depending on the method used. The tumor stages studied vary, with most cases being at the most advanced stages (III and IV). Most of the materials used are derived from formalin-fixed, paraffin-embedded (FFPE) samples, with partial analysis (IHC, tissue microarray (TMA)) or global analysis (whole tissue sections (WTS), flow cytometry, gene expression profiles, and mRNA profiles). IHC sheds light on the heterogeneity between sTILs and iTILs [38]. A study of 37 cases of advanced EOC showed a good correlation between TMA and WTS regarding CD8+ TIL infiltration assessment [39]. CD3+TILs or sTILs [34] and Th17 [40,41] are present at a higher level in EOC than in borderline or benign ovarian. Th1, Th2, and Th17 profiles are diverse within the same patient between tumor and ascites or intra-cystic fluid [29,40,42,43], and between omentum and ovarian tissue [44]. The absolute median count [45] and the CD4/CD8 ratio have a high variability [46]. CD3+T-cells in the ascites increase with a higher frequency of CD4+CD45RA-FoxP3+ T-cells in the ascites compared to the peripheral blood [47]. The frequency of TREG increases in the ovarian tumors compared to the blood samples [48]. Heterogeneity is also seen between tumor grades, with TREG and CD8+ TILs being higher in grade 2 or 3 than in grade 1 [49]. Tertiary lymphoid structures (TLS) are present in HGSOC [50], with a potential role in immunosuppression. Figure 2A is a simplified view of the immunologic network at the tumor site.

A new way to classify cancers is based on gene expression profiles. Using this way, in 2011, the Cancer Genome Atlas (TCGA) divided HGSOC into four distinct groups: mesenchymal, immunoreactive, proliferative, and differentiated [51]. The most used algorithms for genetic analysis, ESTIMATE [52] or CIBERSORT [53], have been used in 379 cases of EOC [54], with 22 immune cells being studied, and confirmed the wide variability in TILs between tumors, especially in CD8+ or TFH. Using the same method in a large EOC cohort (n = 2086, stages I to IV), TCGA and CIBERSORT algorithms allowed us to underline the heterogeneity and to highlight a proportion of 12.2% of macrophages, 6.6% of TFH, and 6.3% of memory CD4 T cells among the TILs [55]. TCGA analysis of 3176 EOC samples illustrate, as expected, the heterogeneity between histologic subtypes, especially between high- and low-grade serous carcinomas [56]. Murakami established an IHC classification that distinguishes four subgroups: mesenchymal transition, immune reactive, solid and proliferative, and papilloglandular [57]. The new classification of IHC has been used in 70 ovarian or peritoneal samples and confirms the heterogeneity in sTIL density [58]. In particular, the heterogeneity in TILs is observable in the primary tumor versus recurrence [59]. For example, the median FOXP3 count is higher during recurrence than at diagnosis [60]. Thus, new techniques are used, including spatioimageomic transcriptomics [61] and imaging mass cytometry +/− combined with machine learning approaches [62], to specify the phenotypic and spatial heterogeneities in TILs in EOC.

The specific markers provide the knowledge on TILs infiltration in EOC, including the type of T cells, cytotoxic T cells, and suppressors. The clinical cohorts used only partially reflect the disease. Indeed, most data are analyzed based on operable diseases from diagnosis, which unfortunately constitute only a part of the patients in practice.

**Table 1 cancers-14-05332-t001:** TIL phenotypes in the reviewed studies.

Study Author (Publication Year)	Number of Cases n=	Tumor Stage	Moment	Subtype/ TIL Phenotype	Specimen Processing	Location	TIL Description
Hagemann (2011) [63]	10	IIIC	pCRS (n = 9) and recurrence (n = 1)	CD3, CD8, FoxP3	primary tumor and two intraperitoneal metastases	iTILs and sTILs	heterogeneity in TIL density inter- and intra-patient (primary *versus* metastasis)
Murakami (2016) [57]	132	I to IV	at diagnosis: pCRS	CD8	tumor samples	iTILs and sTILs	NEW PATHOLOGICAL CLASSIFICATION: mesenchymal transition, immunereactive, solid and proliferative, and papilloglandular
Ojalvo (2017) [64]	52	II to IV	pCRS (37) and recurrence (15)	CD8, FoxP3	tumor samples	iTILs	median FOXP3 count recurrent > primary
Zhu (2017) [41]	126	I to IV	pCRS	CD4, IL17, FoxP3, CD31	ovarian samples of the central areas of EOC	iTILs	% of Treg cells, Th17 cells, and ratio of Treg/Th17 cells: high in patients with EOC
Nakamura (2019) [65]	839	no data	no data	Th1, Th2, Th9, Th17, M1, and M2 macrophage	tumor samples	TILs	Higher intratumoral expressionmarkers may rescue or neutralize the negative associations of inflammation or angiogenesis
Jiménez (2020) [27]	50	IIIC and IV	40 NACT and 10 pCRS	CD45, CD3, CD4, CD8, NK, FoxP3	tumor samples	iTILs	transcriptomic heterogeneity in each patient
Dötzer (2019) [30]	49	IIIC and IV	pCRS (35 CC0, 8 CC1, 6 CC2)	CD45, CD3, CD8, PD-1, PD-L1	tumor sample, peritoneum, and omentum	iTILs and sTILs	differences in the expression between primary cancer and omental and peritoneal lesions
Oberg (2020) [43]	29	III and IV	at diagnosis	CD45, CD8, CD56, CD3, IFN-𝛾, IL-4, IL-9, IL-10, IL-17, TNF-𝛼	blood, ascites, and tumor samples	TILs	heterogeneity in TILs vs ascites
Gao (2020) [55]	2086	I to IV	no data	22 immune cells	tumor samples	TILs	Heterogeneous immune microenvironment: infiltration varied between clinicopathological subgroups (stage, type, and survival)
Lakis (2020) [58]	70	III and IV	pCRS	*H*&*E*, morphological subtypes: IR, SD, PG, MT	tumor samples	iTILs and sTILs	higher sTIL density in implants than in ovarian tissue; heterogeneity between implants of the same patient
Zhou (2021) [54]	379	II to IV	no data	22 immune cells	tumor samples	TILs	heterogeneity in CD8 and NK cells; TFH, monocyte, macrophage: proportion higher in OC than normal tissue
Zhu (2021) [62]	41	IIIB and IV	pCRS	CD8+, CD4+	tumor samples	iTILs and sTILs	heterogeneity between stromal and tumoral tissues
Karakaya (2021) [38]	66	I to IV	at diagnosis	PD-1, CD8, CD4, CD3	tumor samples	iTILs and sTILs	heterogeneity between stromal and tumoral tissues and between histologic types

This table summarizes the studies describing TILs in EOC. The description of the clinical cohort (number of cases, tumor stage, and moment), the type of T cells studied, the tissue specimen, and the location (stroma or intraepithelial) is correlated with the main outcome about TIL description for each of the articles. pCRS: primary cytoreductive surgery, iCRS: intervalle cytoreductive surgery, iTILs: intra-tumoral TILs, sTILs: stromal TILs, NACT: neoadjuvant chemotherapy, OC: ovarian cancer, vs: versus, IR: immune reactive, SD: solid and proliferative, PG: papilloglandular, MT: mesenchymal transition.

### 3.4. TILs and Patients’ Survival

The survival benefits of TILs in EOC have been noted for a long time. In 2003, Coukos and colleagues analyzed 186 advanced EOC tissue samples and detected intra-epithelial CD3+ (iCD3) TILs in 55% of the patients. The 5-year survival rate of these patients was 38%, compared to 4.5% in patients with no detectable TILs [66]. Figure 3 summarizes the studies published in the last ten years that evaluated the impact of TILs on survival. Table 2 shows the characteristics of the patients in the different studies. In the absence of specific antibodies in IHC, most studies employed the H&E analysis [57,67,68,69] with median count or semi-quantitative categories, and survival rates were evaluated (OS and DFS). OS is calculated from the date of histological diagnosis to death (or, in rare cases, from the date of first treatment to the date of death), and DFS is calculated from the date of beginning of treatment to the date of progression or death.

In 2016, Murakami et al. confirmed the positive effect on OS and DFS; the immunoreactive subtype is the group with the better prognosis compared to three of the other groups [57]. TILs are known to be associated with favorable prognostic factors in many solid tumors, including HGSOC [25]. Different types of infiltrating immune cells have varying effects on the prognosis of the patients [36]. CD8+ TILs [66,70,71], Th1 TILs, and Th17 TILs [65,72] are associated with a positive effect. Th2 TILs are associated with either a negative [29,56] or a positive impact [65]. TREG TILs are associated with either a negative [73] or a positive impact [74,75]. CD3-staining iTILs based on the TILs count are counted either manually or using digital imaging analysis to determine the number of T cells per HPF (ranging from 15 to 20 HPF). Several studies found a positive effect of CD3+ TILs [44,63,76,77,78,79], and one showed no impact on the prognosis [80]. The CD3 location can modify the prognosis: sCD3+ TILs are associated with an improved 10-year survival rate [81], whereas iCD3+ TILs have no impact on OS or DFS [82]. Moreover, the studies’ results are sometimes contrasting, with an effect on DFS but not on OS [83], or with a positive effect on OS only [84]. In one study, at diagnosis, most patients showed stromal CD3+ immune infiltration with high heterogeneity in the intra-epithelial CD3+ [85]. In this cohort, while an increase in stromal CD8+/FoxP3+ over 10-fold was associated with a better OS, no association was observed when considering iTILs counts [85].

CD4+ memory TILs (CD45RO+) are most frequently positively correlated with DFS or OS [56,79,86]. This is the case for iCD4+ TILs, but not sCD4+ TILs [87]. However, this positive impact is not consistent [80,82,88]. A significant infiltration of CD8 TILs is most often associated with a positive prognostic effect on OS [24,54,56,76,77,79,81,89,90,91,92]. When only intraepithelial CD8 cells are studied, the results are more heterogeneous, with one study having a positive effect on OS [24], two studies having a positive effect on DFS [87,93], three studies without a significant impact [39,94,95], and three studies having a negative effect on survival [82,96,97]. The results for TREG also vary, with either studies showing a negative impact on OS and DFS [45,78,90,94], or finding no impact or [49,79,82] even a positive impact [45,98].

Some studies evaluated B-TILs and TILs and showed a positive impact on survival [50,81,88,89,99], while others showed no impact at all [82,100]. Using machine learning-based refined differential gene expression and marker combination analysis, 44 markers were evaluated in 839 patients. A higher expression of Th1, Th2, and Th17 associated genes was correlated with better survival outcomes [65]. A 2086 SOC cohort showed that TFH had a negative impact on prognosis [55]. In HGSOC, the presence of CD103 with CD3+ and CD4+ was correlated with a better survival [97,101].

Altogether, recent studies are more controversial than the original study. The variability in the results can be explained by OC heterogeneity, different histology, and intra-patient variability. The clinical cohorts used are also an explanation. Indeed, iCRS are not represented in recent studies (Table 2), and the complete surgery rate is unfrequently used in favor of the optimal surgery rate, which is sometimes much lower than the recommendations for good practice. These confounding factors, type of sample used, and homogeneity of the clinical cohorts reduce the comparability between the studies.

**Table 2 cancers-14-05332-t002:** Patient characteristics in studies evaluating prognostic impact.

Study Author (Publication Year)	Number of Cases n=	Advanced Stage (%)	Serous Histology (%)	Moment	Median Follow-Up (Months)	Optimal Debulking (Residual Tumor <2.5 cm) (%)
Nielsen (2012) [89]	264	92.5	100	pCRS	1.9	30
Bachmayr-Heyda (2013) [93]	203	95.5	88.2	pCRS	48	69.5
Webb (2014) [101]	497	58.9	44.3	pCRS	no data	100
Hermans (2014) [90]	210	100	77.1	At diagnosis	no data	30
De Leeuw (2015) [91]	187	34	100	pCRS	no data	100
Murakami (2016) [57]	132	82	100	pCRS	no data	no data
Santoiemma (2016) [81]	135	71	65.9	no data	no data	59.3
Lundgren (2016) [100]	154	no data	58.4	pCRS	87	no data
Goode (2017) [24]	5078	47.8	62.9	pCRS	48.9	42.4
James (2017) [67]	707	40	44.4	pCRS	no data	no data
Pinto (2018) [87]	128	80.5	100	At diagnosis	no data	80
Hwang (2019) [68]	256	62	56.6	At diagnosis	no data	91
Martin de la Fuente (2020) [84]	130	100	100	At diagnosis	no data	60
Paijens (2021) [97]	268	94.8	100	pCRS (47%) or iCRS	no data	75.7 (51.4 complete)
Wu (2021) [102]	441	87	no data	no data	no data	62.4
Li (2021) [103]	308	88.9	no data	no data	no data	61.4 (18.2 complete)
Chen (2020) [104]	189	59.3	12.7	At diagnosis	37	no data

This table details the clinical characteristics in studies evaluating the prognostic impact of TILs in EOC. Management of EOC is pCRS or iCRS when primary surgery is not possible. Optimally, the goal of surgery is complete resection, with residual tumor: 0 mm. pCRS: primary cytoreductive surgery, iCRS: intervalle cytoreductive surgery, and NACT: neoadjuvant chemotherapy.

### 3.5. Influence of DNA Repair Deficiency and TILs

Genetic alterations in ovarian cancer are dependent on diverse genes. Mismatch repair (MMR) pathway dysregulation represents 5 to 13 % [105] of SOC (including Lynch Syndrome), and BRCA mutations and alterations in homologous recombination are present in 23% and 50%, respectively [106]. Dysregulation of genes implicated in DNA repair leads to a higher mutational burden in ovarian cancers [107], and patients with homologous recombination deficiency (HRD) have been reported to have a higher expression of neoantigens [108,109]. Wang et al. reported a study using TCGA database to analyze the infiltration pattern in ovarian cancer. They classified the tumors into two clusters. The cluster enriched in cytotoxic and immunosuppressive cells tended to have a higher mutational load than the cluster with less immune cell infiltration [110]. Likewise, several studies showed a significantly higher number of CD3+ and CD8+ TILs in HRD [109,111,112] and microsatellite instability tumors [105,113]. There seems to be no difference in TIL infiltrations rates between BRCA1- and BRCA2-mutated patients [109,114]. However, it is noteworthy that homologous recombination proficient (HRP) patients are a heterogeneous group. In this group, some patients express high HLA class 1 molecule and high expression of neoantigens; thus, the accumulation of DNA mutations is not the only process implicated in patients with high immune infiltrate [108]. Understandably, there is more programmed death-ligand (PD-L)1 expression in HRD+ tumors than in HRP when considering combined positive score (CPS); however, interestingly, PD-L1 expression in tumoral cells seems to be equal in the two groups [109]. To conclude, a hypothesis is that high neoantigen load leads to the recruitment of TILs, which is counterbalanced by the expression of immune checkpoints. Lastly, it has been reported that some ovarian cancers with DNA repair deficiency escape immune surveillance despite being a «hot» phenotype according to the morphological diversification of the tumors. This mechanism is being evaluated in 514 cases of advanced HGSOC using TCGA analysis [115]. The ESTIMATE and ABSOLUTE algorithm applications highlight the spatial heterogeneity [115]. Anti-PARPs provide new options for patients presenting with a HRD+ tumor. Understanding the TIL landscape for these tumors is very helpful to propose anti-PARPs in combination with immunotherapy.

### 3.6. TILs and Immune Checkpoints

Programmed death-1 (PD-1 or CD279), with the activation-induced expression on T-cells, can bind to its ligands PD-L1 and PD-L2 to decrease the ability of TILs in destroying tumor cells. PD-L1 overexpression is one way for ovarian cancer to escape the immune surveillance [113]. Monoclonal antibodies targeting the immune inhibitory checkpoints, such as PD-1 and PD-L1, have been tested to evaluate the intensity and quality of T-cell activation [116,117]. PD-(L)1 inhibitor has been approved in the treatment of melanoma, non-small cell lung cancer, small cell lung cancer, head and neck squamous cell carcinoma, urothelial carcinoma, renal cell cancer, and cervical cancer [118]. Biomarkers used to guide treatment are PD-L1s evaluated with IHC, tumor mutation burden, or mismatch repair. In the past, PD-L1 tumor staining was widely used to determine the responders in treated patients. Tumor proportion score (TPS) for PD-L1 is representative of the proportion of tumor cells with membranous PD-L1 expression. More recent studies incorporate the presence of PD-1 or PD-L1 in TILs [118]. Combined positive score (CPS) includes the number of tumor cells, lymphocytes, and macrophages stained with PD-L1 divided by the total number of viable tumor cells and then multiplied by 100 [119]. Figure 2B illustrates the main immune checkpoints studied in EOC. It has been developed to better predict the response to immunotherapy. Table 3 shows the scores in the main-reviewed studies. Scoring expression in the immune checkpoint molecules, the definition of positive TILs, and PD-L1 positive tumors are very heterogeneous. Various studies evaluating HGSOC tissue samples showed that high expressions of PD-1 and PD-L1 in tumors are associated with a better DFS [84,120,121,122,123]. Combined analysis of PD-L1 expression in tumors and CD8+ iTILs allows the stratification of patients based on their prognosis: patients with negative PD-L1 expression in tumors and higher numbers of CD8+ iTILs have the longest median OS, while those with positive PD-L1 expression in tumors and lower numbers of CD8+ iTILs have the shortest median OS [124]. The high density of sTILs-PD-L1+ is associated with a favorable prognostic effect on OS [125]. Exhausted status of CD8+ TILs can be assessed by the co-expression of PD-1 and Tim3 and has been linked to poor prognosis [126]. The localization of the studied tissues seems to have an impact on the prognostic value: TILs expressing PD-1 in carcinomatosis tissue are associated with a better OS and PD-L1 expression in peritoneal tissue is negatively correlated with OS [127]. The expressions of other TIL immune checkpoints in OC, such as T cell immunoglobulin, mucin domain-containing protein 3 (Tim-3), lymphocyte activating gene 3 (LAG-3), and cytotoxic T-lymphocyte-associated protein 4 (CTLA4), have also been evaluated [123]. These checkpoints are also the different targets for other ICIs. Blocking CTLA4, for example, activates CD8+ and CD4+ T cells and enhances the anti-tumor effect of drugs [116]. Targeting Tim-3 is tested in association with anti PD-1 [128]. LAG3 expression has been found to be associated with PD-L1 expression (intra-tumor PD-L1 and CPS score ≥ 1) in 48 HGSOC patients [114], which could lead to combination therapy targeting PD-L1 and LAG3 together.

The contrast in the IC study results can be explained by sampling variation, differences in staining protocols, variability in cut-off values, and variability in the localization of tumors or TILs. TPS and CPS for PD-L1 scoring are approved by the Food and Drug Administration. Immune-checkpoint analysis, particularly TILs, is very likely to be integrated in the future to screen OC patients who might benefit from targeted therapy. A precise description of the immune checkpoints presents in the TILs and on the tumor cells is certainly useful in future trials of ICIs in EOC aiming to appropriately stratify the patients.

### 3.7. Influence of First-Line Chemotherapy on TIL Landscape

We reviewed above the prognostic significance of TILs reported by several studies, most of which were performed on patients undergoing pCRS, as shown in Table 2. Yet, most patients receive platinum-based neoadjuvant chemotherapy (NACT) before iCRS. Depending on the studies, the proportion of CD3+ cells before and after NACT could either be identical [129,130], decreased [131], or increased [69,85]. One explanation for these results is the high inter-patient and intra-patient variability [27]. A way to highlight this heterogeneity is to compare site-matched metastases before and after NACT. A transcriptomic study of 38 matched samples showed an oncogenic expression profile before NACT that evolved to an immune expression profile during chemotherapy. The results showed an increase in the proportion of NK cells and cytotoxic gene set expression after NACT, with no difference in the other subpopulations of TILs. T-cell receptor (TCR) sequencing showed expansion of oligoclonal TCR after NACT in site-matched samples. Other studies showed an increase in CD8+/CD4+ and CD8+/FoxP3+ ratios after NACT [85,132]. These results suggest an anti-tumoral response to chemotherapy with recruitment of NK cells, a decrease in regulatory cells, activation of cytotoxic response, and clonal expansion of T cells.

The prognostic significance of TIL evolution with NACT was explored in 54 patients with advanced ovarian cancer [131]: TIL subpopulations were studied before and after NACT in patients with a good Chemotherapy Response Score (CRS) compared to patients with a poor CRS [133]. Their results showed no differences in immune infiltration density between good and poor responders before NACT. After NACT, there was a decrease in sFoxP3+ cells in good responders, but no difference was seen in poor responders. Good responders showed an increase in IFN-𝛾 expression and a gene expression profile of Th1 activation. There was also an increase in PDL1 expression after NACT in both groups, as reported elsewhere [134]. The results also showed an anti-tumoral response to chemotherapy and a reaction of tumoral cells by expressing PDL1. Therefore, immunotherapy could be more efficient after the first line of chemotherapy.

Most studies involved sample analysis during pCRS. TILs variation that is compared between pre- and post-chemotherapy differs among the studies. It has already been shown that chemotherapy has the potential to alter immunotherapy response [135,136]. The proportion of CD3 TILs does not change after NACT [129,130], whereas NACT induces a decrease in the density of sCD3 TILs in HGSOC patients [131], or an increase in the proportions of sCD3, sCD8, and iCD8 TILs [69,137]. sTILs are associated with platinum sensitivity in 70 patients with advanced-stage SOC [58]. Chemotherapy induces an upregulation of PD-L1 [69,130]. In recurrent HGSOC, a higher density of TILs and a higher expression of MHC have been reported when compared to paired primary tumors, suggesting a higher immunogenicity [85]. The repertoire of neo-epitope recognizing T-cells and their avidity are also different between blood and tumor samples in recurrent disease [138]. The study of TIL evolution, from state of activation to neo-epitope repertoire, under chemotherapy pressure could be key to the development of new personalized immunotherapy.

## 4. Conclusions

Even though immunotherapy is less efficient in EOC compared to other solid tumors, ongoing trials are evaluating the efficacy of combining standard treatments with immunotherapy to improve patients’ prognosis [139,140]. Treatment options that are being evaluated include associating TILs with chemotherapy, antiangiogenic drugs, PARP inhibitors, vaccines, cytokine injections, CAR-T cells, checkpoint inhibitors [141], and even radiotherapy [139,142]. Indeed, combinations of anti-PARP treatment and reactivation of the immune system via anti-PD-L1, PD-L2, or CTLA4 are among the therapeutic options tested in EOC. Standardizing TIL evaluation methods, techniques, and cut-offs is mandatory and is being evaluated in ongoing studies. The biggest challenge now is to harmonize TIL count and immune checkpoint scoring to help develop care strategy. Future personalized medicine will most certainly use the description of the tumor microenvironment, including TILs. 

## Figures and Tables

**Figure 1 cancers-14-05332-f001:**
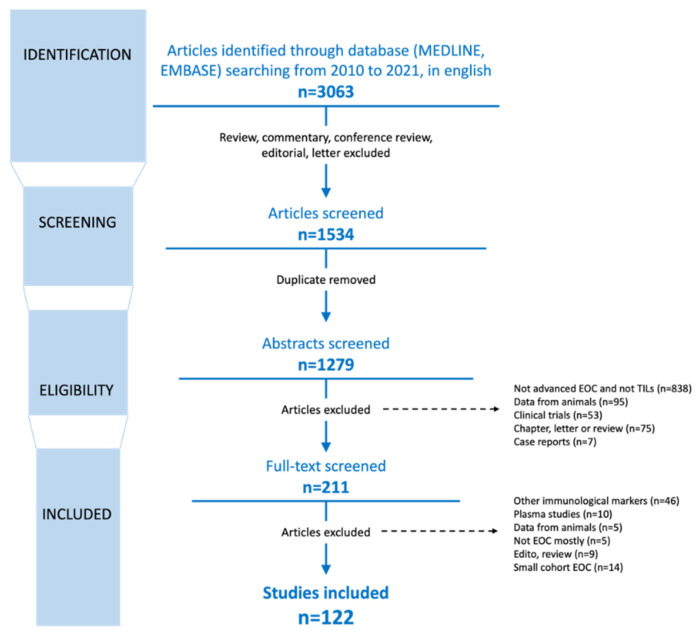
Schematic diagram of the selection process for the studies included in this review. Review according to Moher [21], EOC: epithelial ovarian cancer.

**Figure 2 cancers-14-05332-f002:**
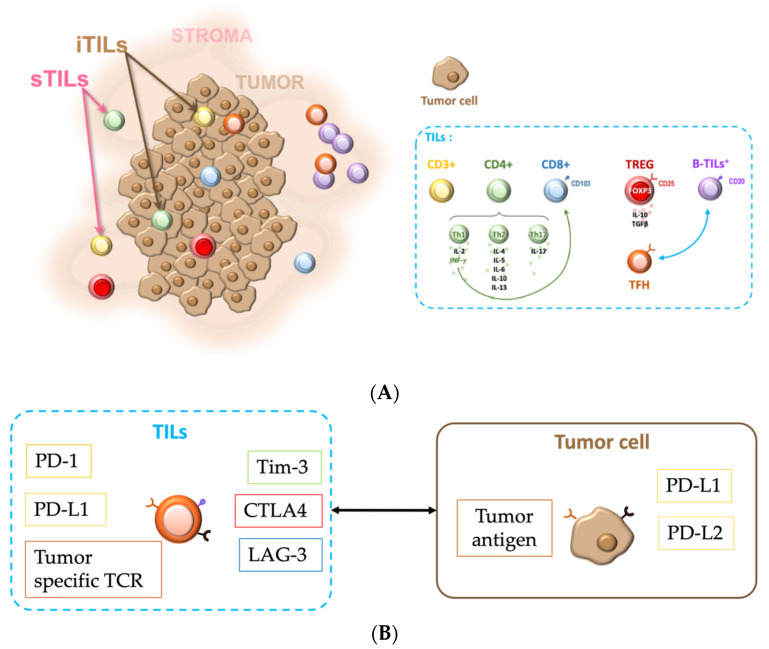
Immunologic network in EOC*: (***A***)* simplified TIL view and location*;*
*(***B***)* Main immune checkpoints studied in EOC. Simplified diagram of the main TILs described in the articles studied in this review. T cells infiltrating the stroma or tumor epithelium are identified via CD3, and/or CD4 and CD8. The subtypes of T cells, including TH1, TH2, TH17, TFH, and TREG, are illustrated. The main immune checkpoints described in this review are represented. iTILs: intra-tumoral, sTILs: stromal, B-TILs: B tumor-infiltrating lymphocytes, TCR: T cell receptor, PD-1: programmed-death 1, PD-L1: PD-1 ligand 1, PD-L2: PD-1 ligand 2, CTLA4: cytotoxic T-lymphocyte-associated protein 4, Tim-3: T cell immunoglobulin and mucin domain-containing protein 3, and LAG-3: lymphocyte activating gene 3.

**Figure 3 cancers-14-05332-f003:**
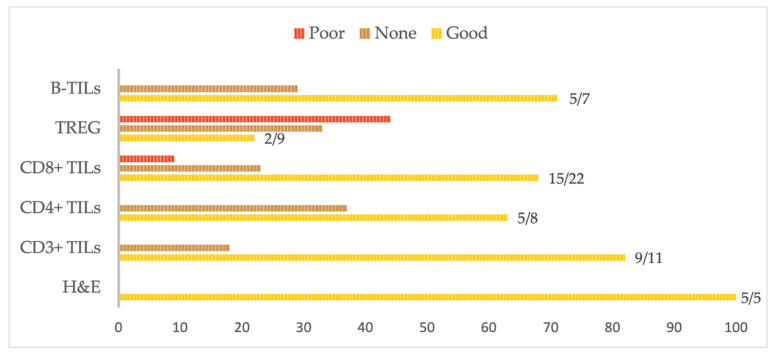
Synthesis of the effects of TILs on prognosis in EOC. This figure summarizes the conclusion of the articles exploring the effects of TILs on EOC prognosis, either being evaluated in HES or via the study of a surface marker. B-TILs are mostly identified using CD20. H&E: hematoxylin and eosin, and B-TILs: B tumor-infiltrating lymphocytes.

**Table 3 cancers-14-05332-t003:** TILs and immune checkpoints in the reviewed studies.

Study Author (Publication Year)	Number of Cases n=	PD-1	PD-L1	Definition of Positive TILs	Scoring PD-1	Scoring PD-L1	Method
Webb (2015) [120]	489	PD-1+ cells: positive factor DFS in HGSOC, and not in other EOCs	Not studied	CD3 and CD8 >5 and < 5	absolute numbers of PD-1+ and CD103+	Not studied	immunohistochemistry, flow cytometry
Webb (2016) [121]	490	Not studied	PD-L1 expression: positive factor OS in HGSOC, and no difference in other EOCs	CD8 quantitative pathology imaging system	Not studied	PD-L1 scored as positive or negative, using a threshold of ≥1 positive cells	immunohistochemistry, TCGA
Darb Esfahni (2016) [122]	215	PD-1 on cancer cells and TILs: positive factor OS	PD-L1 on cancer cells and TILs: positive factor DFS and OS	i and sCD3 cut-off: > 65/mm^2^	cancer cells PD-1+ > 11/mm^2^	cancer cells PD-L1+ > 20/mm^2^	Tissue microarrays + MRNA expression + MA
Wang (2017) [124]	107	Not studied	tumor PD-L1 expression: negative factor for OS; TILs-PD-L1+: no difference for OS	sTILs: score 1 (≤5/HPF), 2 (6–20/HPF), and 3 (≥20/HPF); iTILs: score 1 (≤5/HPF) and 2 (>5/HPF)	Not studied	PD-L1 staining in tumor cells scored: 0, negative; 1, weak expression; 2, moderate expression but weaker than placenta; and 3, equivalent or stronger expression than placenta.	immunohistochemistry
Fucikova (2019) [123]	80	PD-1 high: positive factor OS	PD-L1 (positive vs negative): positive factor OS	CD8: entire TME: absolute number of positive cells/mm^2^; CD20: cell surface/tumor section surface	PD-1, CTLA4, LAG-3: stroma and tumor of whole tumor	PD-L1 intratumoral and stromal, categorized as 1 (0%), 2 (1–4%), 4 (5–9%), and 4 (>10%); cut-off 5% to survival analysis	immunohistochemistry + flow cytometry + TCGA
Kim (2019) [125]	248	Not studied	sTILs-PD-L1+: positive factor OS	stromal sTILs, and iTILs: graded on a semiquantitative scale of 0 (none), 1+ (mild), 2+ (moderate), and 3+ (marked)	Not studied	intensities of PD-L1: intraepithelial (staining in tumor cells)	immunohistochemistry
Martin de la Fuente (2020) [84]	130	high PD-1 expression: better OS	high PD-L1 expression: better OS	grading CD3: 0%, < 1%, 1%, 2–4%, and ≥ 5% (high expression = ≥ 50% cores with ≥ 2% lymphocyte)	PD-1 expression ≥ 1% in ≤50% cores considered high expression	grading PD-L1 and PD-L1: 0%, <1%, 1–4%, ≥5%; iTILs PD-1	Tissue microarray construction and immunohistochemistry
Chen (2020) [104]	189	Not studied	in HGSOC TPS: better DFS and OS (CPS: no difference)		Not studied	TPS and CPS; TPS and CPS ranged from 0 to 100; cutoff score ≥1% for TPS and ≥1 for CPS used to define PD-L1 positivity; for CPS intratumoral and peritumoral, stromal immune cells excluded	immunohistochemistry
Bekos (2021) [127]	111	PD-1 in TILs in peritoneal metastases: positive factor OS	PD-L1 in TILs in peritoneal metastases: negative factor OS	CD8+: % (cut-off 44.3%)	PD-1 % in TILs (cut-off 40%)	PD-L1 % in TILs: ovarian tissue and peritoneal samples (cut-off 15%)	immunohistochemistry
Bansal (2021) [69]	100	Not studied	CPS or TCS PD-L1: no correlation with DFS	sTILs and iTILs: quantified in 5 different (400×) HPF on 3 sections/cases, 0: no lymphoid cells, 1: mild, 2: moderate, and 3: numerous numbers	Not studied	CPS	immunohistochemistry

This review brings together the articles about PD-1 and PD-L1, two preferential ICI targets in EOC. The cut-off, the way of estimating the markers’ expression, and the method used are identified for each study. TPS: tumor proportion score: % tumor cells with membranous PD-L1 expression; CPS: combined positive score: (tumor cell, lymphocyte, and macrophage PD-L1 staining cells)/(total number of viable tumor cells) × 100; EOC: epithelial ovarian cancer; HGSOC high-grade serous ovarian cancer; MA: multi analyze; HPF: high power fields; DFS: disease free survival; and OS: overall survival.

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
