# Peer review of "Tumor-Infiltrating Lymphocytes (TILs) in Epithelial Ovarian Cancer: Heterogeneity, Prognostic Impact, and Relationship with Immune Checkpoints"

_cancers, 2022, doi:10.3390/cancers14215332_

Round 1
Reviewer 1 Report
Review Report
The authors tried to present the role of TILs and their use in ovarian cancer.
The article seemed like a systematic review but later the story is diluted with the introduction of DNA repair and immune checkpoints.
The review is missing the essential statistical analysis showing the relevance of TILs under each subtitle mentioned. They have presented only the outcome of the results.
This review does not give any outcome that is synthesized from the study,
Illustrations to explain the story are not convincing.
Why CD8+ are said to have a positive effect compared to Th2 or TREGs is not explained
How do the different methods of quantification such as manual count and H&E affect the interpretation of these TILs, because H&E is not very specific to the cell type, therefore, it is hard to make any convincing story with these drawbacks.
The legend for the figures is not detailed.
Reviewer 2 Report
Authors performed a systematic literature search to evaluate the available literature (last ten years) and review the roles of the tumor infiltrating lymphocytes (TILs) in epithelial ovarian cancer.
TILs consist of all lymphocytic cell populations that have invaded the tumor tissue. TILs have been described in a number of solid tumors, including ovarian cancer, and are emerging as an important biomarker in predicting the efficacy and outcome of treatment so the information provided by this manuscript will help set the stage for future approaches to optimize clinical utilization of TIL analysis in patients with ovarian cancer.
Although author performed the systematic review search, the study had not been written in a systematic format. Some sentences (258-259; 268-273; 304; 313-317; 384-385; 399-402 and 404-415) need to be described more clearly.
Also, the paper should be carefully revised by a native English speaker /a professional language editing service to improve the readability.
Round 2
Reviewer 1 Report
Accept in present form